# Efficacy and Safety of a 0.1% Tacrolimus Nasal Ointment as a Treatment for Epistaxis in Hereditary Hemorrhagic Telangiectasia: A Double-Blind, Randomized, Placebo-Controlled, Multicenter Trial

**DOI:** 10.3390/jcm9051262

**Published:** 2020-04-26

**Authors:** Sophie Dupuis-Girod, Anne-Emmanuelle Fargeton, Vincent Grobost, Sophie Rivière, Marjolaine Beaudoin, Evelyne Decullier, Lorraine Bernard, Valentine Bréant, Bettina Colombet, Pierre Philouze, Sabine Bailly, Frédéric Faure, Ruben Hermann

**Affiliations:** 1Hospices Civils de Lyon, Hôpital Femme-Mère-Enfants, Centre de Référence pour la Maladie de Rendu-Osler, F-69677 Bron, France; anne-emmanuelle.fargeton@chu-lyon.fr (A.-E.F.); marjolaine.beaudoin@chu-lyon.fr (M.B.); 2Inserm, CEA, Laboratory Biology of Cancer and Infection, University Grenoble Alpes, F-38000 Grenoble, France; sabine.bailly@cea.fr; 3Service de Médecine Interne, CHU Estaing, 63100 Clermont-Ferrand, France; vgrobost@chu-clermontferrand.fr; 4Service de Médecine Interne CHU de Montpellier, Hôpital St Eloi, and Centre d’Investigation Clinique, Inserm, CIC 1411, F-34295 Montpellier CEDEX 7, France; s-riviere@chu-montpellier.fr; 5Hospices Civils de Lyon, Pôle Santé Publique, F-69003 Lyon, France; evelyne.decullier@chu-lyon.fr (E.D.); lorraine.bernard01@chu-lyon.fr (L.B.); 6EA 4129, Faculté de Médecine, Université Lyon 1, 69008 Lyon, France; 7Hospices Civils de Lyon, Pharmacie, Hôpital Louis Pradel, F-69677 Bron, France; valentine.breant@chu-lyon.fr (V.B.); bettina.colombet@ghnd.fr (B.C.); 8Hospices Civils de Lyon, Hôpital de la Croix Rousse, Service ORL, F-69317 Lyon, France; pierre.philouze@chu-lyon.fr; 9Hospices Civils de Lyon, Hôpital E. Herriot, Service ORL, F-69437 Lyon, France; frederic.faure@chu-lyon.fr (F.F.); ruben.hermann@chu-lyon.fr (R.H.)

**Keywords:** hereditary hemorrhagic telangiectasia, epistaxis, nosebleeds, tacrolimus, nasal ointment, genetic disease, rare disease

## Abstract

Hereditary hemorrhagic telangiectasia is a rare but ubiquitous genetic disease. Epistaxis is the most frequent and life-threatening manifestation and tacrolimus, an immunosuppressive agent, appears to be an interesting new treatment option because of its anti-angiogenic properties. Our objective was to evaluate, six weeks after the end of the treatment, the efficacy on the duration of nosebleeds of tacrolimus nasal ointment, administered for six weeks to patients with hereditary hemorrhagic telangiectasia complicated by nosebleeds, and we performed a prospective, multicenter, randomized, placebo-controlled, double-blinded, ratio 1:1 phase II study. Patients were recruited from three French Hereditary Hemorrhagic Telangiectasia (HHT) centers between May 2017 and August 2018, with a six-week follow-up, and we included people aged over 18 years, diagnosed with hereditary hemorrhagic telangiectasia and epistaxis (total duration > 30 min/6 weeks prior to inclusion). Tacrolimus ointment 0.1% was self-administered by the patients twice daily. About 0.1 g of product was to be administered in each nostril with a cotton swab. A total of 50 patients was randomized and treated. Mean epistaxis duration before and after treatment in the tacrolimus group were 324.64 and 249.14 min, respectively, and in the placebo group 224.69 and 188.14 min, respectively. Epistaxis duration improved in both groups, with no significant difference in our main objective comparing epistaxis before and after treatment (*p =* 0.77); however, there was a significant difference in evolution when comparing epistaxis before and during treatment (*p =* 0.04). Toxicity was low and no severe adverse events were reported. In conclusion, tacrolimus nasal ointment, administered for six weeks, did not improve epistaxis in HHT patients after the end of the treatment. However, the good tolerance, associated with a significant improvement in epistaxis duration during treatment, encouraged us to perform a phase 3 trial on a larger patient population with a main outcome of epistaxis duration during treatment and a longer treatment time.

## 1. Introduction

HHT is a rare but ubiquitous hereditary vascular disease, with an estimated prevalence of 1/5000 to 1/8000. The ENG (endoglin) and ACVRL1 encoding ALK1 (activin receptor-like kinase 1) genes are responsible for 90% of cases of HHT [1]. These genes both intervene in the BMP9/ALK1/ENG/SMAD pathway in endothelial cells, and it has been hypothesized that HHT is related to disequilibrium in the angiogenic balance, resulting from an increase in the factors involved in the activation phase and a decrease in those involved in the maturation phase of angiogenesis [2].

The recognized manifestations of HHT are all due to abnormalities in vascular structure. Lesions may be cutaneous and/or mucosal telangiectases or visceral arteriovenous malformations (AVMs) in the lungs, liver, and central nervous system [3]. Telangiectases and AVMs vary widely between individuals and even within the same family. Nosebleeds are the most frequent complication in HHT and may occur as often as several times per day. They are spontaneous, very variable in time and from one patient to another, but recurrent in 90% of patients. They can be associated with severe anemia in 2–10% of patients, and blood transfusions are required sometimes or regularly (every 2 or 3 weeks) in 2–5% of patients [4]. These nosebleeds thus significantly reduce quality of life [5].

The incomplete and transient efficacy of nasal surgical therapies has inspired a new search for adjuvant medical treatments which would greatly diminish daily iron loss [6]. For this reason, anti-angiogenic treatments such as intra-venous anti-VEGF treatment (bevacizumab) and thalidomide have been evaluated in clinical studies [7,8], but their use is limited to severe forms of the disease. Furthermore, local bevacizumab administration (nasal spray) recently evaluated in 2 phase 2 studies was not efficient [9,10]. We thus decided to investigate the feasibility and efficacy on epistaxis in HHT of other known anti-angiogenic drugs with a possible nasal administration and absorption that would target and reactivate the altered BMP9/ALK1/ENG/SMAD pathway.

Based on our collaboration with Bailly’s group, and on the results obtained by Ruiz et al. [11], working on the repositioning approach developed by screening the libraries of Food and Drug Administration approved drugs that could potentiate the BMP9 signaling response, we concluded that the most promising activating drug was tacrolimus, a potent activator of the BMP9-ALK1-BMPR2-Smad1/5/9 signaling cascade. How tacrolimus activates this pathway is still not completely understood. Tacrolimus (FK506) can bind to FKBP12 (FK-506-binding protein-12), a protein known to interact with the TGF-ß family type I receptors [12]. Tacrolimus binding to FKBP12 leads to FKBP12 dissociation from the type 1 receptors, which can then activate the Smad transcription factors. Alternatively, tacrolimus has also been reported to stimulate endoglin and ALK-1 expression in endothelial cells, and to enhance the TGF-β1/ALK1 signaling pathway and endothelial cell functions such as tubulogenesis and migration [13]. In parallel, preclinical models have shown that injections of tacrolimus decreased the number of retinal arteriovenous malformations induced by BMP9/10-immunodepletion in a mouse HHT model [11]. These results suggest that the mechanism of action of FK506 involves a partial correction of endoglin and ALK1 haplosufficiency, and may therefore be an interesting drug for use in patients with HHT. Furthermore, improvement in epistaxis has been shown in HHT patients after a liver transplant [14], and it has been hypothesized that the immunosuppressive treatment (FK506) used to prevent rejection may have an anti-angiogenic effect.

Topical nasal administration of tacrolimus may be an easy-to-use and non-invasive treatment. In addition, tacrolimus ointment is available on the market as a treatment for eczema and can therefore readily be used for nasal administration. Its tolerability on mucosae has been evaluated in the treatment of chronic plaque psoriasis and oral lichen planus [15,16,17,18]. Transient burning sensations have been described in tacrolimus patient groups, but no serious adverse effects necessitating stopping treatment have been recorded. Furthermore, none of the patients showed any abnormality in hematological or biochemical parameters. Data from healthy human subjects indicate that there is little or no systemic exposure to tacrolimus following repeated topical application of tacrolimus ointment. Several studies in infants [19,20] on the pharmacokinetics of tacrolimus after first and repeated application showed minimal systemic exposure (less than 1 ng per mL in all cases) and there was no evidence of systemic accumulation. Further to administration on mucosae, the pharmacokinetics were evaluated in two clinical trials and no systemic absorption was detected [16,18].

For all these reasons, tacrolimus ointment was a good candidate treatment for HHT. Our objective was to evaluate the efficacy on the duration of nosebleeds of tacrolimus nasal ointment in patients with hereditary hemorrhagic telangiectasia complicated by nosebleeds.

## 2. Materials and Methods

### 2.1. Trial Design and Treatment

The study was a prospective, phase II multicenter, randomized study, ratio 1:1, carried out in a double-blind setting. It was approved by the local research ethics committee and by the French Medical Products Agency (ANSM) in March 2017. Written informed consent was obtained from all patients in accordance with national regulations. The trial was conducted in accordance with the principles of the Declaration of Helsinki [21] and Good Clinical Practice guidelines. All the authors were involved in designing or conducting the study, and preparing the manuscript, including the decision to submit it for publication. This trial was registered with the ClinicalTrials.gov Identifier #NCT03152019.

The nasal ointment was self-administered by patients, twice daily, for 6 weeks. About 0.1 g of product was to be administered in each nostril. The tube was gently squeezed to extract an amount roughly equivalent to the size of the head of a cotton swab. The ointment was then introduced into each nostril with a cotton swab, and extended into the nostril with a cotton swab or/and by external pressure on the nostril.

Marketed 30 g tubes of Protopic^®^ 0.1% (Léo Pharma, Voisins le Bretonneux, France) were used for this study. For the placebo, the manufacturing and filling were managed by an external pharmaceutical laboratory with GMP accreditation. The placebo formulation was similar to the active ointment; it contained all the ingredients except the active one: tacrolimus. The product was provided in strictly identical tubes. The same masked label was placed on both batches in order to respect the blind.

### 2.2. Participants

This study enrolled patients over the age of 18 years, with clinically confirmed HHT suffering from epistaxis (more than 30 min during the 6 weeks prior to the time of inclusion justified by completed follow-up grids), and who had not undergone nasal surgery in the 6 weeks prior to inclusion. We did not include women who were pregnant or those likely to become so during the study, or patients with known hypersensitivity to macrolides in general, to tacrolimus or to any of the excipients, or patients who had incompletely filled in the nosebleed grids, or patients with an inherited skin barrier, or with CYP3A4 inhibitor treatment (erythromycin, itraconazole, ketoconazole, and diltiazem), or patients with ongoing immunosuppressive treatment, or with known and symptomatic immune deficiency.

### 2.3. Patient Information and Follow-Up

Patients were informed and recruited during a standard consultation with the ENT doctor or doctor responsible in the reference center or skill center for HHT, and informed of the study and the need to complete nosebleed grids for ENT monitoring for the 6 weeks prior to the start of the treatment. Patients were included during a consultation at the reference center or skill center for HHT and the treatment was prescribed during the same consultation.

The follow-up consisted of 2 phone calls on days 15 and 31 (i.e., 14 and 30 days after the beginning of the treatment) in order to collect information regarding tolerance and observance, and in visits (with medical and ENT consultations) at the end of the 6 weeks of treatment, and at 6 weeks after the end of the treatment.

### 2.4. Study End Points

The main outcome was the percentage of patients experiencing an improvement in their nosebleeds. An improvement was defined as a 30% reduction in the total duration of nosebleeds over the 6 weeks following treatment, compared with the duration of the nosebleeds in the 6 weeks before the treatment.

Secondary outcomes were total duration of nosebleeds, number of nosebleeds, and number of red blood cell transfusions before, during, and after treatment, progress in the scores obtained in the SF36 quality of life questionnaire and in the ESS (Epistaxis Severity Score) using data from the specific questionnaire and biological efficacy criteria (hemoglobin and serum ferritin). All were recorded before treatment and at 6 and 12 weeks after the end of the treatment.

### 2.5. Safety

Safety was evaluated at each visit by means of a physical examination (monitoring of blood pressure, clinical ear, nose, and throat examination to check the nasal septum and other side effects on nasal mucosa), and assessment for adverse events (AE). All AE were coded using the Medical Dictionary for Regulatory Activities (MedDRA). Adverse events were classified by the investigators as unrelated, dubitable, or possibly, probably or certainly related to the treatment. Monitoring the safety of administration of the product, motivated by the iatrogenic risks, justified the setting up of a specific independent monitoring and safety committee. The committee met in particular in the case of the occurrence of serious adverse events and gave its recommendations on the continuation of the study after collection of adverse events and observance of the treatment after 30 days for the first 8 patients included. It was composed of a specialist of the disease not involved in the study, an ENT specialist, and a statistician specialized in the methodology of clinical trials.

Systemic absorption of tacrolimus was evaluated by means of FK506 dosages in blood samples 8, 22, and 43 days after the beginning of the treatment. No systemic absorption and effects were expected, but, in case of a positive dosage of 5 ng/mL or more, it was decided that the laboratory would immediately inform the investigator in order to ask the patient to stop the treatment.

### 2.6. Sample Size Calculation

We hypothesized that 60% of patients would be improved in the treatment group against 15% in the placebo group. It was therefore necessary to include 22 patients in each group to reach an 80% power with a 5% alpha (bilateral), leading to 44 patients overall (Fisher exact test).

Taking into account early withdrawal and patients who may be lost to follow-up, we planned to include 24 patients in each group, that is to say, a total of 48 patients.

### 2.7. Randomization

The randomization process was centralized. Patients were randomized by blocks of 4 and unstratified. Allocation of a randomization arm to an included patient was made by IWRS (Interactive Web Response System), on the basis of a unique randomization list for all investigation centers. The randomization list was pre-established, by the “Pole IMER” at the Hospices Civils de Lyon–Clinical Research Unit. Clinsight software version 7.1 (Ennov Clinical^®^, Paris, France) was used to manage this study. After verifying the inclusion criteria, the investigator connected to the platform to create the list of patients. Once the inclusion criteria had been validated, the patient was randomized and a treatment code was allocated by the system. The treatment was then dispensed by the pharmacy at the Hospital Center. This was a double-blind study in which neither the patient nor the investigator was aware of the nature of the treatment administered.

### 2.8. Statistical Methods

Populations: 2 populations were defined. The per protocol population, which was set at 70% adherence, consisted of all patients receiving at least 60 ointment treatments of the 84 planned. The intention-to-treat (ITT) population consisted of all randomized patients starting the treatment, and patients were considered in their randomization group. All analyses were performed on the ITT populations; the main outcome was also analyzed on the per protocol population.

Initial characteristics of the patients were summarized by means of descriptive statistics (number, average, standard deviation, median, minimum, and maximum for the quantitative variables, and numbers and percentages for the qualitative variables).

Analysis of the main outcome: the percentage of patients experiencing improvement in their nosebleeds was computed in each group. The percentage was compared between groups using a Chi^2^ test (or Fisher exact test if the conditions for Chi^2^ were not fulfilled), and the analysis was performed on the intention-to-treat population and on the per protocol population. Patients who stopped the treatment but who had filled in epistaxis grids were analyzed using their data. Patients who withdraw from the study before completing the follow-up were considered as failures.

Analysis of the secondary outcomes: the percentage of patients with at least one adverse event was computed and compared between the 2 groups. Quantitative parameters were presented as mean ± standard deviation and median (minimum and maximum) for all groups and were compared using the Student *t*-test (or Mann–Whitney test in case of non-normality). Qualitative parameters at inclusion were presented in terms of number (percentage) and compared using the Chi^2^ test (or Fisher exact test where conditions for the Chi^2^ test were not fulfilled). Mixed models were produced to compare evolution between the groups.

All analyses were performed using SAS software version 9.4 (SAS Institute Inc., Cary, NC, USA). Effect sizes were computed as risk difference (Chan–Zhan 95% CI) and relative risks for binary outcomes, and as Cohen’s d for quantitative outcomes.

### 2.9. Missing Data

The main outcome was based on grids that were filled in daily. If one day was missing, the value was replaced by an average of the 4 values before and the 4 values after the missing value. This strategy was applied up to 7 missing values over 6 weeks (i.e., 10%). If more than 7 days and less than 21 days (included) were missing, a daily average was computed from the data available (from the 6-week period evaluated) and multiplied by 42 to estimate epistaxis duration. If a patient was lost to follow-up or refused to communicate his nosebleed grids or had more than 21 days missing on his grids, the result for the patient concerned was considered as a failure.

## 3. Results

### 3.1. Trial Population

After screening 155 patients, 50 patients were included and randomized between May 2017 and August 2018 in three different centers (Lyon, Clermont-Ferrand, and Montpellier). The baseline characteristics are summarized in Table 1. All individuals except one met the inclusion criteria and were enrolled in the study. Due to one wrong allocation in the placebo group and to one wrong inclusion, the independent committee recommended that it was necessary to include two more patients (50 instead of the 48 patients initially scheduled) (Figure 1). One patient in the placebo group discontinued his follow-up due to a severe adverse event. All patients in the tacrolimus group filled in the epistaxis grids and completed six weeks of treatment. Twenty-five patients in the placebo group filled in the epistaxis grids and out of them 24 completed six weeks of placebo treatment.

### 3.2. Response to Treatment

#### 3.2.1. Primary Outcome

All patients in the intention-to-treat population were analyzed (*n* = 50). The results are summarized in Table 2. According to our main outcome (30% reduction in the total duration of epistaxis over six weeks after treatment), no statistical difference was observed in the tacrolimus groups compared to the placebo group. Analysis of the per protocol population led to the same conclusions.

#### 3.2.2. Secondary Outcomes

Duration and number of nosebleeds before, during, and after treatment are presented in Figure 2. As for percentage of improvement, there was no difference between the tacrolimus group and the placebo group regarding evolution in the parameters during the six weeks following treatment. However, there was a difference in evolution when comparing epistaxis before and during treatment in terms of epistaxis duration (*p =* 0.04, Cohen’s d: 0.53 (–0.04–1.11) and epistaxis number (*p =* 0.04, Cohen’s d: 0.39 (−0.19–0.96)) (Figure 2).

The number of red blood cell transfusions did not differ before and during treatment (*p =* 0.57, Cohen’s d: −0.21 (−0.77–0.36)), or before and after treatment (*p =* 0.69, Cohen’s d: −0.23 (−0.8–0.34)). Three patients had blood transfusions before treatment and all of them were in the placebo group. Of them, two patients also had blood transfusions during treatment.

The SF-36 questionnaire revealed no differences in the dimensions of quality of life before, during, and after treatment (Table 3).

The difference in the ESS after and before treatment in the tacrolimus group (mean = –0.43 (SD = 1.47)) and the placebo group (mean = −0.26 (SD = 0.99)) (*p =* 0.69, Cohen’s d: 0.13 (–0.53–0.8)), and during and before treatment in the tacrolimus group (mean = -1.47 (SD = 1.55)) and the placebo group (−0.96 (SD = 1.26)) (*p =* 0.31, Cohen’s d: 0.36 (−0.35–1.07)) were not significantly different.

The biological criteria (hemoglobin and ferritin levels) did not significantly improve during or after treatment. The effect sizes for evolution in these parameters were 0.38 (−0.2–0.95) and 0.11 (−0.46–0.69) during treatment, and were even lower after treatment (0.19 (−0.38–0.76) and −0.32 (−0.9–0.26)), respectively. Mean levels on inclusion, six weeks after the beginning of the treatment, and six weeks after the end of the treatment in both the tacrolimus and placebo groups were 12.8, 13.3, 13.0 and 12.6, 12.6, and 12.6 for hemoglobin (g/dL), and 51.3, 75.6, 43.9 and 49.0, 59.9, and 69.3 for ferritin (µg/L), respectively. Using mixed models, we did not find a significant trend over time for hemoglobin levels (*p =* 0.61) or ferritin levels (*p =* 0.36), and no difference in evolution between groups was observed (*p =* 0.59 for hemoglobin and 0.60 for ferritin).

### 3.3. Safety Outcomes

A total of 51 AE (25 in the tacrolimus group and 26 in the placebo group) and three severe adverse events (SAE) (one in the tacrolimus group and two in the placebo group) were recorded without differences between the groups. No SAE certainly or probably related to the treatment were recorded.

Of the 29 possibly or probably related AE, 13 patients had nose burning or a tingling sensation (10 in the tacrolimus group and three in the placebo group), two had infection, both in the tacrolimus group (genital HSV infection (*n* = 1), intercostal VZV infection (*n* = 1)), seven had a local sensation (burning eyes (*n* = 1), burning throat (*n* = 2), sneezing (*n* = 1), rhinitis (*n* = 1) and nose smell (*n* = 2) (four in the tacrolimus group and three in the placebo group) and five had other symptoms (thoracic pain (*n* = 1), back pain (*n* = 2), diarrhea (*n* = 1) and headache (*n* = 1), all in the placebo group.

Systemic absorption of tacrolimus: as expected, all FK506 dosages were < 5.0 ng/mL on day 8 (*n* = 48), day 22 (*n* = 46) and day 43 (*n* = 46) after the beginning of the treatment. Tacrolimus was detected in two patients (4.3%), both in the tacrolimus group on day 8 (1.2 and 1.02 ng/mL), four patients (8.3%) on day 22 (1.03, 1.02, 1.2 and 1.05 ng/mL), and not detected on day 43 (0%).

## 4. Discussion

To our knowledge, this was the first phase II, double-blind, multicenter, randomized, placebo-controlled trial evaluating the efficacy of tacrolimus nasal ointment on epistaxis in HHT. In the present study, there was no significant benefit to using tacrolimus ointment administered twice a day on the nasal mucosa for six weeks after the end of the treatment, compared with placebo. We chose this time point for our main outcome (before treatment vs. after the end of the treatment) on the basis of physiology and on our previous studies using anti-angiogenic drugs which usually improved patients many weeks after the beginning of the treatment. Tacrolimus enhanced the ALK-1 signaling pathway in the endothelial cells of HHT patients, and inhibited increased VEGF signaling and hypervascularization in an HHT animal model [11], and we hypothesized that the effect which involved “blood vessel remodeling” would not be immediate. Furthermore, recent clinical data in one patient treated with low oral doses of tacrolimus showed that ESS improvement was observed three months after the beginning of the treatment [22]. However, we did not take into account that this treatment was local and with good local absorption, as confirmed by blood dosages which were very low but positive in four cases, and maybe had a quick effect during treatment which stopped after the end of the treatment. Moreover, the vast majority of patients in the tacrolimus group had a prior history of nasal surgery, compared to only half in the placebo group. It can be argued that this disequilibrium may have blunted the apparent effect of tacrolimus during the treatment period and/or contributed to the lack of an obvious effect after tacrolimus was halted. Similarly, fewer women were assigned to the tacrolimus than the placebo group, which may have influenced the results.

Importantly, we observed for our secondary outcome a significant improvement in epistaxis duration and number during the treatment phase in the tacrolimus group. These results suggest that the effect of the drug occurred only during treatment and patients relapsed after they stopped the treatment. However, although this trial was randomized, we know that moistening nasal mucosa improves epistaxis in HHT [10], and we cannot exclude the idea that the efficacy observed in the treatment group was partly related to the effect of the ointment, not to the drug itself. Furthermore, this observation highlighted the fact that prolonged tacrolimus use would be necessary, and we do not have data on the long-term safety on mucosae. The ESS did not improve during treatment; however, even though the ESS has been shown to be a validated tool in the evaluation of epistaxis in HHT [23,24], we chose not to use it as our main outcome. This was firstly because our patients are used to completing epistaxis grids, and secondly because the ESS is self-administered by patients and is more subjective than an epistaxis duration measurement. It effectively includes subjective questions such as intensity, need for medical attention and anemia, not defined by hemoglobin level but by our patients’ own evaluation.

Tolerance of the nasal ointment was good after a 6-week treatment. No severe adverse events were observed. The most frequent related adverse event was a sensation of burning in the nose in 34.5% of patients in the tacrolimus group, but it was transient in most cases. None of the patients stopped the treatment for this reason. This result was similar in non-HHT patients receiving tacrolimus ointment on mucosae (oral lichen planus) in randomized studies. Corrocher et al. [16] and Vohra et al. [17] observed a burning sensation in 56% and 35% of patients, respectively, treated for oral lichen planus, and, similarly, this event resolved rapidly within four to five days. In the present study, we observed two infections in the tacrolimus group but not locally, one genital and one skin infection, and the pharmacokinetics monitoring performed revealed moderate systemic absorption, almost undetectable in most cases. It is thus unlikely that these complications were related to an immunosuppressive effect of tacrolimus. Other adverse events were observed with the same frequency in both groups. Nasal cartilaginous septum perforation was followed closely and not observed in either group after treatment.

This trial had several significant limitations. First, patients completed epistaxis grids and noted epistaxis duration, which are not directly observed outcomes, partly subjective and imprecise, and are subject to error. Second, we included all HHT patients with nosebleeds and did not take into account a history of nasal surgery or nasal crusts or septal perforation, which may change mucosal drug absorption. Almost all patients had undergone different types of surgery and most of the mucosa of the nasal cavity cannot be touched by a cotton swab, thus possibly resulting in under treatment. Third, the blinding process was successful in most cases; however, burning sensations were more frequent in the tacrolimus group. This could lead to underestimation of the placebo effect and overestimation of the relative treatment effect because patients can deduce that they are in the placebo arm and may be less likely to report improvement [25]. Finally, we observed a decrease in epistaxis duration in the placebo group, during and after treatment, which could be partly due to a Hawthorne effect. While nasal moisturizing is a recommended treatment for preventing HHT bleeding, participating in a study would probably improve the way it was done.

## 5. Conclusions

In HHT patients with epistaxis, tacrolimus nasal ointment, administered twice daily compared with a placebo, did not reduce monthly epistaxis duration in the six consecutive weeks after treatment compared to the six weeks immediately before the start of treatment. However, good tolerance associated with a significant improvement in epistaxis duration during treatment encouraged us to perform a phase 3 trial on a larger patient population with a main outcome of epistaxis duration during treatment and a longer treatment time.

## Figures and Tables

**Figure 1 jcm-09-01262-f001:**
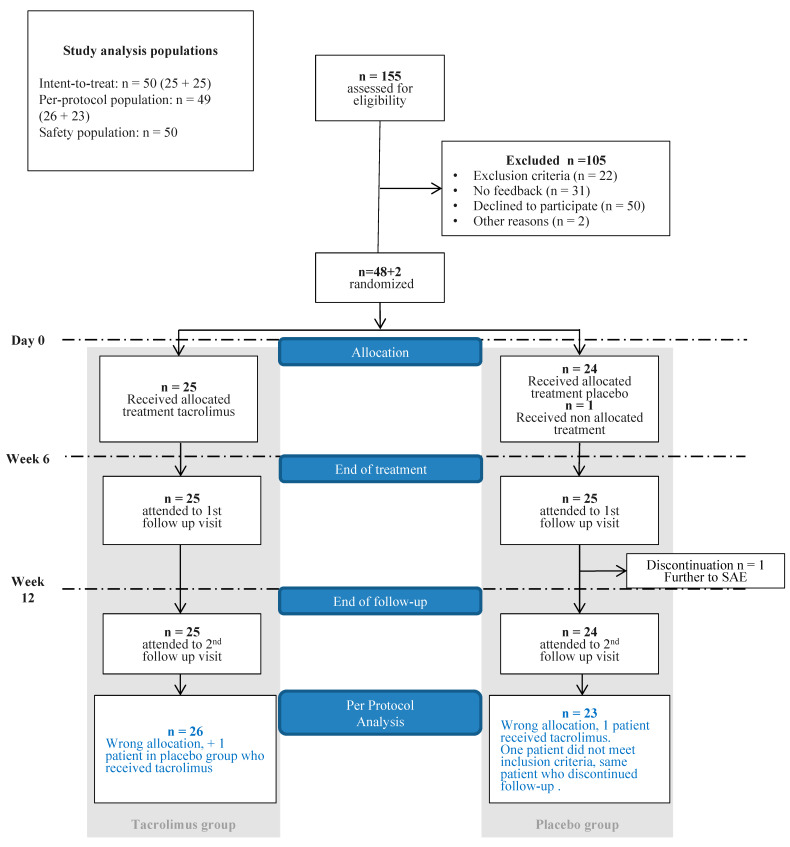
Flow chart.

**Figure 2 jcm-09-01262-f002:**
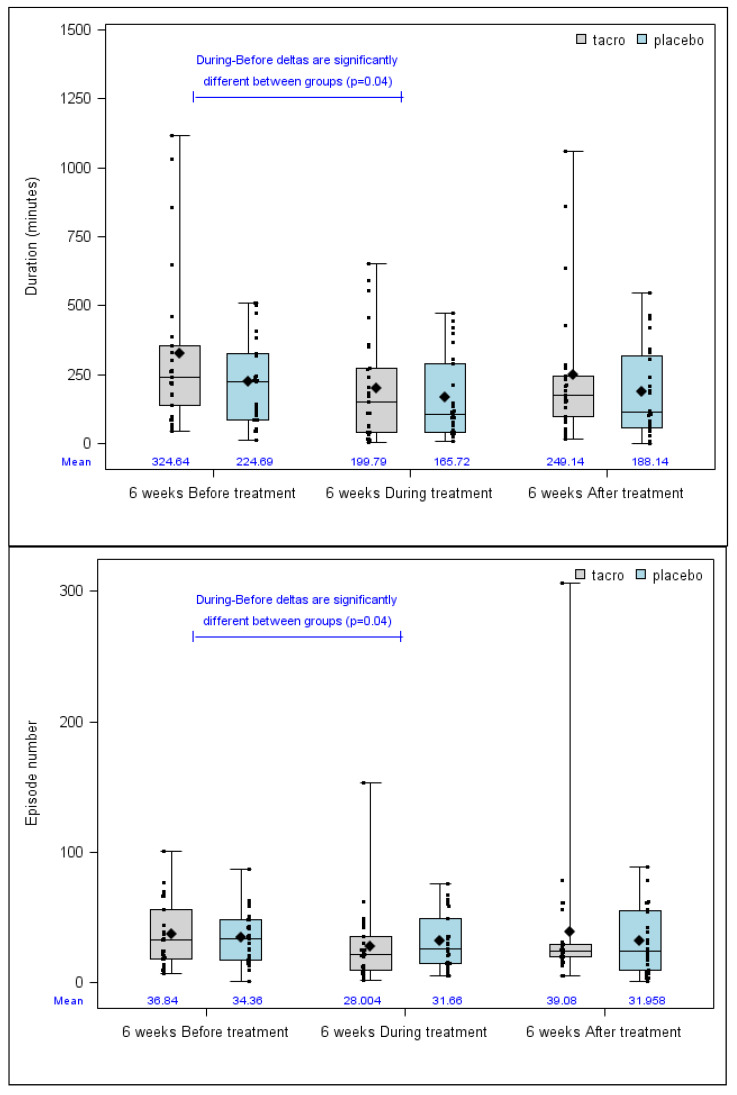
Mean epistaxis duration and number on six weeks before, during and after treatment. Legends: Lines from bottom to top: minimum, 25th percentile, median, 75th percentile and maximum. Diamond: mean. Small squares: individual data.

**Table 1 jcm-09-01262-t001:** Patient characteristics before treatment.

Variable	Modality	All	Tacro Group *n* (%)	Placebo Group *n* (%)
Number	*n*	50	25	25
Age (years)	Median (min–max)	62 (27–85)	60 (27–81)	64 (39–85)
	Mean (SD)	60.92 (11.26)	59.04 (12.26)	62.8 (10.06)
Females	*n* (%)	23 (46)	9 (36)	14 (56)
Mutated gene	*n* (%)			
ALK1		36 (72)	20 (80)	16 (64)
ENG		10 (20)	5 (20)	5 (20)
On-going		2 (4)		2 (8)
Not known		2 (4)		2 (8)
Blood transfusions in the last 6 weeks before inclusion	*n* (%)	2 (4)	0 (0)	2 (8)
Parameters on inclusion (D0)				
Nasal surgery	*n* (%)	35 (70)	21 (84)	14 (56)
Nasal septum perforation	*n* (%)	7 (14.3)	3 (12.5)	4 (16)
Hemoglobin level	Mean ± SD	126.62 (22.66)	127.6 (20.82)	125.64 (24.75)
(g/dL)	Median (Min–Max)	130 (66–163)	129 (90–163)	130 (66–158)
Ferritin level (ng/mL)	Mean ± SD	50.12 (73.7)	51.28 (94.79)	48.96 (45.83)
	Median (Min–Max)	28 (4–458)	23 (4–458)	33 (6–174)
Systolic blood pressure (mmHg)	Mean ± SD	130.7 (20.98)	133.16 (16.5)	128.24 (24.78)
	Median (Min–Max)	126.5 (100–181)	130 (110–163)	124 (100–181)
Diastolic blood pressure (mmHg)	Mean ± SD	80.24 (16.32)	82.2 (11.42)	78.28 (20.13)
	Median (Min–Max)	80 (28–129)	80 (60–105)	80 (28–129)

**Table 2 jcm-09-01262-t002:** Main outcome (*n* = 50) analysis: Efficacy of tacrolimus ointment on mean epistaxis duration six weeks before and after treatment.

Variable	Modality	All	Tacro Group *n* (%)	Placebo Group *n* (%)	*p*-Value	Effect Size *
Epistaxis duration decrease > 30% (ITT)	No	31 (62)	15 (60)	16 (64)	0.77	RD 4.0 (−23.4–31.2) RR 1.11 (0.55–2.26)
	Yes	19 (38)	10 (40)	9 (36)		
Epistaxis duration decrease > 30% (PP)	No	30 (61.2)	16 (61.5)	14 (60.9)	0.96	RD −1.0 (–28.2–28.0) RR 0.98 (0.49–1.99)
	Yes	19 (38.8)	10 (38.5)	9 (39.1)		
**Other data related to primary outcome**						
Epistaxis total duration 6 weeks before treatment (min)	*n*	50	25	25	0.34	−0.42 (–1.00–0.15)
	median (min–max)	226.5 (11–1116)	240 (46–1116)	226 (11–510.8)		
	Mean (SD)	274.67 (239.24)	324.64 (292.06)	224.69 (162.35)		
Epistaxis total duration 6 weeks immediately after the end of the treatment (min)	*n*	49	25	24	0.42	−0.29 (−0.86–0.29)
	median (min–max)	170 (1–1058)	177 (16–1058)	114.5 (1–547)		
	Mean (SD)	219.26 (213.72)	249.14 (252.7)	188.14 (163.43)		

Legends: * risk difference (RD) and relative risks (RR) for binary outcomes; and Cohen’s d for quantitatives.

**Table 3 jcm-09-01262-t003:** Evolution of SF36 scores before and during treatment (B/D) and before and after treatment (B/A).

Variable			All	Tacro Group	Placebo Group	*p*-Value *	Effect Size **
Physical functioning	B/D	*n*	46	22	24	0.89	0.04 (−0.55–0.63)
		median (min–max)	0 (−25–40)	0 (−15–20)	0 (−25–40)		
		Mean (SD)	0.43 (11.44)	0.68 (9.55)	0.21 (13.14)		
	B/A	*n*	49	25	24	0.38•	0.07 (−0.5–0.64)
		median (min–max)	0 (−35–55)	0 (−35–20)	0 (−30–55)		
		Mean (SD)	−0.51 (14.62)	0 (10.99)	−1.04 (17.88)		
Physical role	B/D	*n*	45	22	23	0.28•	0.39 (−0.21–0.99)
		median (min–max)	0 (−50–125)	0 (−50–125)	0 (−50–75)		
		Mean (SD)	5 (37.91)	12.5 (42.08)	−2.17 (32.78)		
	B/A	*n*	46	23	23	0.78•	0.3 (−0.29–0.9)
		median (min–max)	0 (−75–125)	0 (−50–125)	0 (−75–50)		
		Mean (SD)	2.17 (35.68)	7.61 (40.19)	−3.26 (30.44)		
Bodily pain	B/D	*n*	46	22	24	0.53	0.19 (−0.41–0.78)
		median (min–max)	0 (−59–38)	0 (−59–38)	0 (−29–22)		
		Mean (SD)	−0.02 (17.36)	1.68 (20.24)	−1.58 (14.51)		
	B/A	*n*	49	25	24	0.88•	−0.12 (−0.69–0.45)
		median (min–max)	0 (−43–22)	0 (−38–22)	0 (−43–22)		
		Mean (SD)	−0.49 (15.66)	−1.4 (17.14)	0.46 (14.26)		
General health	B/D	*n*	46	22	24	0.36	0.27 (−0.32–0.87)
		median (min–max)	0 (−20–25)	2.75 (−20–23.75)	0 (−20–25)		
		Mean (SD)	2.06 (10.06)	3.5 (11.35)	0.74 (8.75)		
	B/A	*n*	49	25	24	0.59	0.15 (−0.42–0.73)
		median (min–max)	0 (−17–28.25)	0 (−15–28.25)	0 (−17–25)		
		Mean (SD)	0.14 (10.01)	0.9 (10.23)	−0.65 (9.93)		
Vitality	B/D	*n*	46	22	24	0.45	0.23 (−0.37–0.82)
		median (min–max)	0 (−25–30)	3.34 (−10–25)	0 (−25–30)		
		Mean (SD)	2.32 (11.81)	3.71 (9.64)	1.04 (13.59)		
	B/A	*n*	49	25	24	0.64	−0.13 (−0.71–0.44)
		median (min–max)	−3.33 (−35–35)	−3.33 (−35–35)	−2.5 (−20–20)		
		Mean (SD)	−2.11 (12.54)	−2.93 (14.06)	−1.25 (10.96)		
Social functioning	B/D	*n*	46	22	24	0.67	0.13 (−0.47–0.72)
		median (min–max)	0 (−37.5–37.5)	12.5 (−25–37.5)	0 (−37.5–37.5)		
		Mean (SD)	5.16 (16.37)	6.25 (15.79)	4.17 (17.16)		
	B/A	*n*	49	25	24	0.93	0.02 (−0.55–0.6)
		median (min–max)	0 (−25–37.5)	0 (−25–37.5)	0 (−25–37.5)		
		Mean (SD)	2.81 (16.39)	3 (16.65)	2.6 (16.48)		
Emotional role	B/D	*n*	46	22	24	0.69•	0.21 (−0.38–0.8)
		median (min–max)	0 (−100–100)	0 (−66.67–100)	0 (−100–100)		
		Mean (SD)	5.8 (43.49)	10.61 (44.11)	1.39 (43.38)		
	B/A	*n*	49	25	24	0.42•	0.18 (−0.39–0.75)
		median (min–max)	0 (−66.67–100)	0 (−66.67–100)	0 (−33.33–100)		
		Mean (SD)	7.48 (35.53)	10.67 (41.63)	4.17 (28.34)		
Mental health	B/D	*n*	46	22	24	0.1	−0.5 (−1.1–0.1)
		median (min–max)	1 (−19–28)	0 (−19–24)	4 (−16–28)		
		Mean (SD)	2.59 (9.9)	0.05 (9.54)	4.92 (9.83)		
	B/A	*n*	49	25	24	0.03•	−0.65 (−1.24–0.06)
		median (min–max)	0 (−44–24)	0 (−44–24)	8 (−12–24)		
			1.33 (11.68)	−2.24 (13.27)	5.04 (8.52)		

* Student *t*-test (or Mann–Whitney in case of non-normality, signaled by “•”) ** Effect size: Cohen’s d.

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
