# Peer review of "Efficacy and Safety of a 0.1% Tacrolimus Nasal Ointment as a Treatment for Epistaxis in Hereditary Hemorrhagic Telangiectasia: A Double-Blind, Randomized, Placebo-Controlled, Multicenter Trial"

_jcm, 2020, doi:10.3390/jcm9051262_

Round 1

Reviewer 1 Report

The authors are commended for performing a clinical trial in this needy population. The authors have shown that the tacrolimus treatment, as administered in this study, did not have tangible benefit beyond the nasal lubrication provided by the ointment, as demonstrated by the control group.

The authors have acknowledged several weaknesses in their trial design. However, the authors should consider adding to this section the method by which the nose was treated with tacrolimus. The amount that was applied, and the method by which it was applied would serve to treat only a minimal portion of the nasal mucosa; the majority of which is transformed and can bleed in patients with HHT. Further, there were several patients in the study whom had a septal perforation. Given the location of a septal perforation, it is difficult to know where the nasal ointment was actually applied in the nose for those affected patients, given the method employed to apply the nasal "treatment" in this trial. It should be realised the mucosa of the nasal cavity occupies far more area than what can be touched by a cotton swab, thereby inadvertently under-treating the mucosa of the majority of the nasal cavity in the trial. If the tacrolimus had been applied more widely in the nasal cavity, using a different method of application, then the purported benefits of the medication might have been better appreciated. 

Author Response

Question 1: the authors should consider adding to this section the method by which the nose was treated with tacrolimus. The amount that was applied, and the method by which it was applied would serve to treat only a minimal portion of the nasal mucosa; the majority of which is transformed and can bleed in patients with HHT.

Response 1: We added to the design section the method by which the nose was treated which was detailed later: “The nasal ointment was self-administered by patients, twice daily, for 6 weeks. About 0.1 g of product was to be administered in each nostril. The tube was gently squeezed to extract an amount roughly equivalent to the size of the head of a cotton swab. The ointment was then introduced into each nostril with a cotton swab, and extended into the nostril with a cotton swab or/and by external pressure on the nostril.

Marketed 30 g tubes of Protopic® 0.1% were used for this study. For the placebo, the manufacturing and filling was managed by an external pharmaceutical laboratory with GMP agreement. The placebo formulation was similar to the active ointment; it contained all the ingredients except the active one: tacrolimus. The product was provided in strictly identical tubes. The same masked label was placed on both batches in order to respect the blind.”

Question 2: Further, there were several patients in the study whom had a septal perforation. Given the location of a septal perforation, it is difficult to know where the nasal ointment was actually applied in the nose for those affected patients, given the method employed to apply the nasal "treatment" in this trial. It should be realised the mucosa of the nasal cavity occupies far more area than what can be touched by a cotton swab, thereby inadvertently under-treating the mucosa of the majority of the nasal cavity in the trial. If the tacrolimus had been applied more widely in the nasal cavity, using a different method of application, then the purported benefits of the medication might have been better appreciated. 

Response 2: We thank reviewer 1 for this comment and have added this point to the discussion, line 388-391: “Second, we included all HHT patients with nosebleeds and did not take into account a history of nasal surgery or nasal crusts or septal perforation, which may change mucosal drug absorption. Almost all patients had undergone different types of surgery and most of the mucosa of the nasal cavity cannot be touched by a cotton swab, thus possibly resulting in under-treatment.”

Reviewer 2 Report

SUMMARY

In this study, authors conduct the first prospective, double-blinded, placebo-controlled phase II study of the efficacy of tacrolimus, an FDA-approved BMP9 signaling cascade activator, to treat epistaxis in HHT patients. Authors find that tacrolimus did not significantly improve epistaxis duration 6 weeks after conclusion of treatment, but that epistaxis duration and number was modestly improved during the 6-week treatment period. In addition, authors present evidence that the drug is relatively well tolerated by patients, rationalizing progression to a phase III study that is currently underway.

MAJOR CONCERNS

Based on preclinical studies of tacrolimus in BMP9/10-deficient mice, tacrolimus has been proposed as a promising new drug for HHT treatment. Therefore, this study addressing the efficacy of tacrolimus in human HHT patients is important, and impactful for HHT researchers and patients. Moreover, this phase II study is for the most part well-designed and well-executed, and many of the limitations in the study’s design (i.e. the duration of tacrolimus treatment) would not have been obvious before the outcomes reported herein. Overall, this study is scientifically sound and findings – in regards to both the apparent efficacy of tacrolimus as well as indications of its safety in this patient population -- are interesting and generally merit publication.

However, this manuscript suffers from a rather weak discussion section, and would be improved by a more detailed discussion of the study’s design and limitations particularly with regard to the chosen treatment frequency and duration, as well as expanded consideration of other issues only briefly referenced in the final paragraph of the Discussion section. There is existing evidence that an intranasal saline spray alone improves epistaxis in HHT (reference 10, Whitehead et al. 2016), but authors only briefly address the possibility that their findings may be related to simple nasal moistening. In addition, authors’ finding that tacrolimus appears ineffective following cessation of treatment warrants consideration of the long-term safety of prolonged tacrolimus use, as well as comparison to bevacizumab and thalidomide efficacy.  

Authors could also add further consideration to rule out possible confounding factors in their dataset. First, authors mention that they did not address patient history of nasal surgery, which could have contributed to differences in epistaxis severity between the two groups, and may also have confounded the degree of mucosal drug absorption in the tacrolimus treatment group. Table 1 shows that the vast majority of patients in the tacrolimus group had prior history of nasal surgery, compared to only half of the placebo group – could this have blunted the apparent effect of tacrolimus during the treatment period and/or contributed to the lack of an obvious effect after tacrolimus was halted?

Similarly, Table 1 shows that fewer women were assigned to the tacrolimus group than the placebo group. A recent study (Mora-Lujan et al 2020) suggests gender differences in the severity of visceral involvement in HHT (although that study also reported no difference in ESS). Does patient gender influence ESS and/or tacrolimus efficacy in this study?

MINOR CONCERNS

  • Citation formatting was lost throughout the Discussion section and should be corrected.
  • It would be helpful to include in brackets that the per protocol population was set at 70% adherence (vs. merely saying 60 of 84 treatments).
  • Line 236 “to obtain the overall estimated duration” was unclear upon first reading and should instead “estimated epistaxis duration”.
  • Figure 1 contains red underlines that should be corrected.
  • Lines 379-380 regarding nasal moistening: sentence construction – “participating in a study would probably improve the way doing it” -- is odd and the meaning is unclear.

Author Response

MAJOR CONCERNS

Question 1: This manuscript suffers from a rather weak discussion section, and would be improved by a more detailed discussion of the study’s design and limitations particularly with regard to the chosen treatment frequency and duration, as well as expanded consideration of other issues only briefly referenced in the final paragraph of the Discussion section. There is existing evidence that an intranasal saline spray alone improves epistaxis in HHT (reference 10, Whitehead et al. 2016), but authors only briefly address the possibility that their findings may be related to simple nasal moistening. In addition, authors’ finding that tacrolimus appears ineffective following cessation of treatment warrants consideration of the long-term safety of prolonged tacrolimus use, as well as comparison to bevacizumab and thalidomide efficacy.  

Response 1: We thank the reviewer for these important comments and we have added this sentence to the discussion, paragraph 2 (lines 364-368): “However, although this trial was randomized, we know that moistening nasal mucosa improves epistaxis in HHT[10] and we cannot exclude the idea that the efficacy observed in the treatment group was partly related to the effect of the ointment, not to the drug itself. Furthermore, this observation highlighted the fact that prolonged tacrolimus use would be necessary, and we do not have data on the long-term safety on mucosae.”

Question 2: Authors could also add further consideration to rule out possible confounding factors in their dataset. First, authors mention that they did not address patient history of nasal surgery, which could have contributed to differences in epistaxis severity between the two groups, and may also have confounded the degree of mucosal drug absorption in the tacrolimus treatment group. Table 1 shows that the vast majority of patients in the tacrolimus group had prior history of nasal surgery, compared to only half of the placebo group – could this have blunted the apparent effect of tacrolimus during the treatment period and/or contributed to the lack of an obvious effect after tacrolimus was halted?

Response 2: That is an important point which has been added to the discussion (lines 359-362): “Moreover, the vast majority of patients in the tacrolimus group had a prior history of nasal surgery, compared to only half in the placebo group. It can be argued that this disequilibrium may have blunted the apparent effect of tacrolimus during the treatment period and/or contributed to the lack of an obvious effect after tacrolimus was halted”.

Question 3: Table 1 shows that fewer women were assigned to the tacrolimus group than the placebo group. A recent study (Mora-Lujan et al 2020) suggests gender differences in the severity of visceral involvement in HHT (although that study also reported no difference in ESS). Does patient gender influence ESS and/or tacrolimus efficacy in this study?

Response 3: We thank the reviewer for this comment. However, as underlined by the reviewer, there were no differences in male vs female populations in the ESS in the Moran-Lujan study or epistaxis duration in other studies we have performed. We have added a sentence, line 363: “Similarly, fewer women were assigned to the tacrolimus than the placebo group which may have influenced the results.”

MINOR CONCERNS

Question 4: Citation formatting was lost throughout the Discussion section and should be corrected.

Response 4: citation formatting has been updated.

Question 5: It would be helpful to include in brackets that the per protocol population was set at 70% adherence (vs. merely saying 60 of 84 treatments).

Response 5: This point has been added, line 221: “The per protocol population, which was set at 70% adherence, consisted of all patients receiving at least 60 ointment treatments of the 84 planned”.

Question 6: Line 236 “to obtain the overall estimated duration” was unclear upon first reading and should instead “estimated epistaxis duration”.

Response 6: The sentence has been modified, line 251: “If more than 7 days and less than 21 days (included) were missing, a daily average was computed from the data available (from the 6-week period evaluated) and multiplied by 42 to estimate epistaxis duration”.

Question 7: Figure 1 contains red underlines that should be corrected.

Response 7: Figure 1 has been modified and red underlines have been deleted.

Question 8: Lines 379-380 regarding nasal moistening: sentence construction – “participating in a study would probably improve the way doing it” -- is odd and the meaning is unclear.

Response 8: The sentence has been modified: Line 405-406 “…participating in a study would probably improve the way it was done”
